# Hemispheric Differences of 1 Hz rTMS over Motor and Premotor Cortex in Modulation of Neural Processing and Hand Function

**DOI:** 10.3390/brainsci13050752

**Published:** 2023-05-02

**Authors:** Jitka Veldema, Dennis Alexander Nowak, Kathrin Bösl, Alireza Gharabaghi

**Affiliations:** 1Department of Sport Science, Bielefeld University, 33615 Bielefeld, Germany; 2Department of Neurology, VAMED Hospital Kipfenberg, 85110 Kipfenberg, Germany; 3Department of Neurology, University Hospital Marburg, 35043 Marburg, Germany; 4Institute for Neuromodulation and Neurotechnology, University Hospital and University of Tübingen, 72076 Tübingen, Germany

**Keywords:** rTMS, hand function, primary motor cortex, dorsal premotor cortex, laterality differences

## Abstract

Introduction: Non-invasive brain stimulation can modulate both neural processing and behavioral performance. Its effects may be influenced by the stimulated area and hemisphere. In this study (EC no. 09083), repetitive transcranial magnetic stimulation (rTMS) was applied to the primary motor cortex (M1) or dorsal premotor cortex (dPMC) of either the right or left hemisphere, while evaluating cortical neurophysiology and hand function. Methods: Fifteen healthy subjects participated in this placebo-controlled crossover study. Four sessions of real 1 Hz rTMS (110% of rMT, 900 pulses) over (i) left M1, (ii) right M1, (iii) left dPMC, (iv) right dPMC, and one session of (v) placebo 1 Hz rTMS (0% of rMT, 900 pulses) over the left M1 were applied in randomized order. Motor function of both hands (Jebsen–Taylor Hand Function Test (JTHFT)) and neural processing within both hemispheres (motor evoked potentials (MEPs), cortical silent period (CSP), and ipsilateral silent period (ISP)) were evaluated prior and after each intervention session. Results: A lengthening of CSP and ISP durations within the right hemisphere was induced by 1 Hz rTMS over both areas and hemispheres. No such intervention-induced neurophysiological changes were detected within the left hemisphere. Regarding JTHFT and MEP, no intervention-induced changes ensued. Changes of hand function correlated with neurophysiological changes within both hemispheres, more often for the left than the right hand. Conclusions: Effects of 1 Hz rTMS can be better captured by neurophysiological than behavioral measures. Hemispheric differences need to be considered for this intervention.

## 1. Introduction

Repetitive transcranial magnetic stimulation (rTMS) possesses the capability to non-invasively modulate specific cortical brain regions. Thus, its application can improve the understanding of the neural background of human behavior and support the development of stimulation protocols for disabled cohorts.

### 1.1. Modulation of Neural Networks and of Hand Motor Function by rTMS

The available data demonstrate that a single session of rTMS can induce neurophysiological changes up to 120 min beyond the stimulation period [1,2], and its persistence increases linearly with the number of pulses applied [1,3,4]. Stimulation frequencies are regarded as the primary factors determining the direction of rTMS-induced changes. A simplified categorization of either “facilitatory” or “inhibitory” protocols has been established over the past years. High-frequency rTMS (≥5 Hz), iTBS, paired pulse rTMS (inter-stimulus interval 1.5 ms), and paired associative rTMS (inter-stimulus interval 25 ms) are expected to induce an up-regulation of neural processing. In contrast, low-frequency rTMS (1 Hz), cTBS, paired-pulse rTMS (inter-stimulus interval 3 ms), and paired associative rTMS (inter-stimulus interval 10 ms) are considered to have down-regulating effects [5]. However, more recent studies demonstrated rTMS-induced effects beyond this traditional view. It has been shown, for example, that several protocols (1 Hz, 10 Hz, 15 Hz, 20 Hz, iTBS, cTBS) can both increase and decrease corticospinal excitability [6,7,8]. The amount and the direction of rTMS-induced changes correlated with the pre-interventional MEP latency [7], the pre-interventional MEP variability, and the late I-wave recruitment [8]. Thus, the variability of neural responses to a single TMS pulse may induce differential neural plasticity, within and across individuals, when applying rTMS. In addition to these factors, sex, age, genetics, and medication may also impact the effects of (r)TMS on neural networks, as detected by a previous systematic review [9] and two recent large-scale analyses [10,11]. Aside from individual factors, also technical (stimulator type, neuro-navigation use, TMS pulse waveform) and methodological (target muscle and hemisphere, time after stimulation, time of day, behavioral context) aspects significantly determine the stimulation-induced effects [9,10,11].

### 1.2. Modulation of Hand Function by rTMS

Present evidence about rTMS-induced modulation of hand function is significantly scarcer than data about rTMS-induced changes of neural networks. The available data indicate that all common stimulation protocols can both support and deteriorate hand motor function in healthy individuals [12,13,14,15,16,17]. Thus, rTMS-induced effects on hand function in healthy individuals are more heterogenous than the traditional view of supportive effects by “facilitatory” and deteriorating effects by “inhibitory” protocols. It is cogitable that the highly inconsistent effects of individual protocols on the neural network [1,6,7] lead to opposite effects on hand function.

### 1.3. Stimulation Location Dependent Effects of rTMS

Previous rTMS research on healthy people focuses almost only on left-hemispheric application [1]. Similarly, the investigations of rTMS-induced neural changes are mostly limited to the left hemisphere [1]. A focus on the dominant hemisphere (and neglecting the non-dominant hemisphere) may generate a bias on understanding the potential of rTMS in the modulation of the neural motor network and motor behavior. Several studies demonstrated hemispheric disparities in both healthy and disabled people.

A recent TMS-study demonstrated that the inhibitory influence from the left to the right hemisphere during active movement of the right hand is higher than the inhibitory influence from the right to left hemisphere during active movement of the left hand [18]. A current analysis of TMS data detected that cortical excitability within motor areas of the left hemisphere was lower than that within the right hemisphere during motor inactivity in right-handed healthy subjects [11]. Lesion studies indicate great relevance of the dominant (left) hemisphere during recovery. Patients with left hemispheric stroke took two to three times longer to learn a movement task with either hand in comparison to patients with right hemispheric damage [19]. Similarly, motor recovery of the affected hand during a three-weeks period was twice as large in patients with right-hemispheric than left-hemispheric stroke [20]. The 1 Hz rTMS over the non-affected hemisphere induced significant improvement of the affected hand in patients with left-hemispheric injury, but not in patients with right-hemispheric damage [20].

Thus, a systematic application of rTMS on either side and investigation of its effects within either hemisphere and hand may foster our understanding of the neurophysiological and behavioral mechanism of hand motor control, and thereby contribute to the development of tailored stimulation protocols. Furthermore, great evidence is desirable for rTMS applications outside M1 regions. More than 80% of existing rTMS studies have focused on M1 and the remaining brain areas have been insufficiently investigated [1]. This, even though existing data indicates that, e.g., premotor and supplementary motor areas or frontal cortex regions are promising targets [1].

We demonstrated in a previous study that 1 Hz rTMS over dPMC is effective in the modulation of hand motor function in stroke subjects [21], and we will continue to investigate this auspicious region. The aim of the current study was to test and compare the effectiveness of 1 Hz rTMS over M1 and dorsal premotor cortex (dPMC) in the left and right hemisphere, in modulation of the neural processes in both M1 and the motor function of both hands in healthy people.

## 2. Methods

### 2.1. Participants

The inclusion criteria were as follows: (1) age over 18 years; (2) no coexistent neurological, psychiatric, or orthopedic illness; (3) no contraindications for (r)TMS [22]. All subjects gave their written informed consent prior to participation. The study was approved by the Ethics committee of the Bavarian chamber of physicians (EC no. 09083).

### 2.2. Study Design

This study was a randomized, placebo-controlled crossover trial. All participants completed five interventional sessions: (1) 1 Hz rTMS over the left M1; (2) 1 Hz rTMS over the right M1; (3) 1 Hz rTMS over the left dPMC; (4) 1 Hz rTMS over the right dPMC; (5) placebo rTMS over the left M1. Simple randomization (using sealed envelopes) was used to determine their order, with a washout period of at least 48 h in between. Prior and after each intervention, neural processing within both hemispheres (corticospinal excitability (motor evoked potentials (MEPs)), long-lasting cortical inhibition (cortical silent period (CSP)) and interhemispheric inhibition (ipsilateral silent period (ISP)) and motor function of both hands (Jebsen–Taylor Hand Function Test (JTHFT)) were evaluated. Subjects were seated in a comfortable chair within a quiet, moderately illuminated room during the experiment.

### 2.3. Evaluations

#### 2.3.1. Neurophysiological Evaluations

A transcranial magnetic stimulator (Magstim Super Rapid; Magstim Co., Dyfed, Whitland, UK) with a 70-mm figure-of-eight coil was used to apply single TMS pulses over each hemisphere separately. The coil was placed tangentially in a posterior–anterior plane at a 45° angle from the midline. Electromyographic (EMG) activity was recorded using silver–silver-chloride electrodes positioned in a belly tendon technique over the FDI muscles. LabChart and PowerLab software were used for data acquisition and analysis (AD-Instruments, Australia).

First, the motor hotspots of both FDI muscles were determined by systematically maneuvering the coil across the scalp. The hotspots are defined as the coil position that resulted in the largest and most reliable MEP amplitude and were recorded from the fully relaxed contralateral FDI muscles. Second, the resting motor threshold was determined for each hotspot as the minimal stimulator output intensity that elicited MEPs with a peak-to-peak amplitude of at least 50 µV from the fully relaxed contralateral FDI muscles in at least five out of ten trials. Third, MEP amplitude, CSP duration, and ISP duration were assessed for each hemisphere. The stimulation intensity for all neurophysiological evaluations was determined during the first pre-interventional examination and kept constant throughout the remaining experiment.

##### MEP Amplitude

The participants were instructed to relax and rest their hands on their lap during the examination. Twenty single TMS pulses at 110% of the resting motor threshold were applied over the hotspot of M1 representing the FDI muscle. Peak-to-peak MEP amplitudes were recorded from the relaxed contralateral FDI muscle.

##### CSP Duration

The participants were instructed to perform an opposition pinch grip at 20–30% of their FDI maximal voluntary isometric contraction capacity (based on maximal voluntary contraction test via a force transducer made before) and maintain a constant level of muscle activity during the test. The applied force was continuously recorded by a force transducer and displayed on a computer screen to ensure the correct force level during the test. Ten single TMS pulses at 130% of the resting motor threshold were applied over the hotspot of the contralateral FDI muscle. The CSP duration was measured from the beginning of the MEP amplitude (onset) to the time of reappearance of an EMG amplitude that was 3-fold the standard deviation of the background EMG noise at rest (offset) [23,24].

##### ISP Duration

The participants were instructed to perform an opposition pinch grip at 90–100% of their FDI maximal voluntary isometric contraction, and maintain a constant level of muscle activity during the test. A force transducer and the procedure described above were used to determine the maximal force level and to maintain the required force during the measurement. Ten single TMS pulses at 150% of the resting motor threshold were applied over the hotspot of the ipsilateral FDI muscle. The onset of ISP was defined as the first suppression of EMG below the lower variation limit (3-fold the standard deviation). The offset was determined as the first return above the lower variation limit [24,25].

#### 2.3.2. Hand Motor Function Evaluation

Motor function of both hands was assessed using the JTHFT [26]. The test consists of six tasks which simulate activities of daily motor performance: turning of cards, picking up small common objects, simulated feeding, stacking checkers, picking up large objects, and picking up large heavy objects. The participants were instructed to complete each task as fast as possible. The time taken for the task performance was recorded using a stopwatch.

### 2.4. Interventions

Using the same magnetic stimulator and a figure-of-eight stimulation coil, 900 pulses of (1) 1 Hz rTMS over left M1, (2) 1 Hz rTMS over right M1, (3) 1 Hz rTMS over left dPMC, (4) 1 Hz rTMS over right dPMC, and (5) placebo rTMS over left M1 were applied. Stimulation intensities of 110% of rMT (real rTMS) or 0% of rMT (with the stimulator output set to 0% for placebo rTMS) were used. The coil was placed tangentially with the handle directed at 45° to the posterior–anterior plane over the targeted area. For stimulation of both M1, the coil was placed over the hotspots of the contralateral FDI muscles. For stimulation of both dPMC, the coil was placed 2 cm anterior and 1 cm medial to the motor hot spots of each hemisphere [27,28]. To control the placement of the coil during the experiment, its exact location was marked on the scalp for both M1 and dPMC using a waterproof felt pen during the first session.

### 2.5. Analysis

The data collected during the experiment was analyzed using the SPSS software package version 27 (International Business Machines Corporation Systems). The pre-interventional comparability was evaluated using independent sample *t*-tests. Repeated measure ANOVAs with factor “intervention” and “time” compared the pre-post changes across interventions. Correlation analyses between the pre-post changes of the behavioral and the neurophysiological parameters were performed using Pearson tests. A *p*-value of ≤0.05 was considered statistically significant.

## 3. Results

In total, 17 participants were randomized. Two participants withdrew from the study because of headache during/after TMS application, and their data is not included. The remaining 15 participants (age 32 ± 11 years, 14 females, one male, 13 right-handers, 2 left-handers (determined by the Edinburgh Handedness Questionnaire [29])) tolerated the interventions well without adverse events. Table 1 presents data on the hand motor function test (JTHFT) and electrophysiological measures (MEP, CSP, ISP) for each intervention and time point (pre, post). The pre-interventional data did not differ significantly across interventions, with two exceptions (see Table 1).

### 3.1. ANOVAs

The ANOVAs detected significant time*intervention interaction on CSP and ISP, but not on MEP and the hand motor test (JTHFT). Stimulation of the right primary motor cortex (F_1,14_ = 6.472; *p* = 0.023) and the left primary motor cortex (F_1,14_ = 5.043; *p* = 0.041) induced a lengthening of CSP duration within the right hemisphere in comparison to placebo stimulation. Similarly, stimulation of the right primary motor cortex (F_1,14_ = 7.605; *p* = 0.015) and the left dorsal premotor cortex (F_1,14_ = 5.459; *p* = 0.035) induced a lengthening of ISP duration within the right hemisphere in comparison to placebo stimulation. Furthermore, stimulation of the right primary motor cortex induced a lengthening of ISP duration within the right hemisphere in comparison to stimulation of left primary motor cortex (F_1,14_ = 5.058; *p* = 0.041). Figure 1 demonstrates intervention-induced changes of hand motor function and electrophysiological measures expressed as absolute differences between pre- and post-evaluation.

### 3.2. Correlations

Significant correlations were found between intervention-induced changes of hand motor function and intervention-induced changes of electrophysiological measures for (1) placebo rTMS, (2) 1 Hz rTMS over left M1, (3) 1 Hz rTMS over right M1, and (4) 1 Hz rTMS over right dPMC (Figure 2).

Left hand: Changes of motor function of the left hand as measured by JTHFT correlated significantly with (i) changes of MEP-amplitude size within the left hemisphere (r = 0.504, *p* = 0.038 (placebo rTMS)) and the right hemisphere (r = −0.602, *p* = 0.018 (placebo rTMS); r = −0.585, *p* = 0.022 (1 Hz rTMS over left M1)), (ii) changes of CSP duration within the left hemisphere (r = −0.579, *p* = 0.024 (1 Hz rTMS over right dPMC)), and (iii) changes of ISP duration within the left hemisphere (r = 0.555, *p* = 0.032 (1 Hz rTMS over right dPMC)). Thus, an improved motor performance (expressed as reduced time to perform JTHFT) was associated with (i) an increase of MEP amplitude within the contralateral hemisphere and a decrease of MEP amplitude within the ipsilateral hemisphere, (ii) a prolongation of CSP duration within the ipsilateral hemisphere, and (iii) a reduction of ISP duration within the ipsilateral hemisphere.

Right hand: Changes of motor function of the right hand correlated significantly with changes of ISP duration within the right hemisphere (r = −0.553, *p* = 0.033 (1 Hz rTMS over right dPMC)). An improved motor performance was associated with a prolongation of ISP duration within the ipsilateral hemisphere.

## 4. Discussion

The aim of this study was to investigate and compare the effect of 1 Hz rTMS over the left and the right M1 and dPMC on (i) motor function of both hands and (ii) neurophysiological processing within both hemispheres. The data indicate that the stimulation protocol is effective to modulate neural processing but not hand motor function. The correlation analysis demonstrated numerous significant relationships between neural and behavioral changes.

### 4.1. rTMS-Induced Motor Effects

Our study did not detect the relevant effects of 1 Hz rTMS on hand motor function, regardless of the area or hemisphere stimulated. Part of prior research shows consistent effects [30,31,32]. However, other studies demonstrated the supportive effects of 1 Hz rTMS over M1 on hand motor function in healthy cohorts [33,34]. A closer look at stimulation protocols indicates that while subthreshold intensities (<100% of rMT) are associated with an improved hand motor function [30,31,32], suprathreshold intensities (>100% of rMT) induce no significant changes of hand motor performance [33,34]. This observation is in accordance with our results. Regarding 1 Hz rTMS over dPMC, only little previous evidence exists [35,36]. The data indicate that this protocol is not effective in modulating hand function in healthy cohorts [35,36], in line with our findings.

### 4.2. rTMS-Induced Changes of Corticospinal Excitability and Relationships to Motor Performance

No significant modulation of corticospinal excitability was detected in our study, regardless of area and hemisphere stimulated. In contrast, previous research shows that 1 Hz rTMS over M1 may induce both inhibition as well as facilitation of corticospinal excitability within the stimulated [1,37,38] and the contralateral hemisphere [1,37]. Applications over dPMC induced either suppression [1,39,40,41,42,43] or no modulation [41,42,43] of corticospinal excitability in healthy cohorts.

Pre-post comparison of our MEP data demonstrates considerably greater standard deviations for all targeted areas in comparison to placebo stimulation (Figure 1). This means that 1 Hz rTMS induced both an increase and a decrease of corticospinal excitability in either hemisphere compared to placebo. In accordance, previous studies demonstrated a great variability of rTMS-induced neural changes for 1 Hz rTMS and several other protocols [6,7].

The correlation-analysis demonstrates that rTMS-induced improvement of left-hand motor function is associated with both an increase of corticospinal excitability within the right hemisphere and a decrease of corticospinal excitability within the left hemisphere. In contrast, an earlier trial indicated that stimulation-induced motor benefits may correlate with both, an increased, and a decreased corticospinal excitability within the contralateral hemisphere [44]. Other studies showed that non-invasive brain stimulation evokes both neural and behavioral changes in healthy cohorts, however, without mutual relationships [45,46].

### 4.3. rTMS-Induced Changes of CSP Duration and Relationships to Motor Performance

Our study demonstrates a lengthening of the CSP duration within the right hemisphere after stimulation of each, the right and left, M1. DPMC targeting induced non-significant prolongation of CSP duration in both hemispheres in comparison to placebo. A big part of previous studies demonstrated a lengthening of CSP latency after 1 Hz rTMS over M1 [1,38,47,48,49,50,51], in accordance with our results. While earlier research showed more pronounced changes within the stimulated hemisphere, our study demonstrated a stronger right-hemispheric modulation. Regarding 1 Hz rTMS over dPMC only slight previous evidence exists [51]. The data indicates that this protocol is not effective in CSP latency modulation [51]. Our data indicate that this protocol is less effective than M1 modulation, however significant effects could be potentially detectable in larger cohorts.

Our correlation approach shows that an improved motor function of the left hand is associated with a prolonged CSP duration within the left hemisphere. A recent study found relationships between stimulation-induced lengthening of the left hemispheric CSP and worsening of the right-hand motor function in healthy probands [15].

### 4.4. rTMS-Induced Changes of ISP and Relationships to Motor Performance

Our data demonstrate significant lengthening of ISP latency in the right hemisphere after 1 Hz rTMS over both right M1 and left dPMC. Right M1 stimulation leads to a stronger lengthening of ISP duration than left M1 targeting. Evident but non-significant prolongation of ISP latency was observed within the left hemisphere after left M1 and right dPMC stimulation. In contrast, previous work demonstrated no effects of 1 Hz rTMS on ISP latency [47,48].

The correlations analysis shows that while an improvement of the left-hand performance correlates with reduced ISP duration within the left hemisphere, an improvement of the right hand is associated with prolonged ISP duration within ipsilateral M1. Previous studies did not detect similar in between hemisphere differences. The available data indicate that favorable performance of either hand is associated with longer ipsilateral ISP [52,53].

### 4.5. Stimulation Location Dependent Effects

Our data demonstrate hemispheric and hand disparities regarding the stimulation-induced effects with 1 Hz rTMS over either hemisphere, inducing significant changes of neural processing within the right M1, but not within the left M1. Our observation receives support from a study that applied 5 Hz rTMS, 10 Hz rTMS, and TBS over supplementary motor cortices of both hemispheres and detected greater suppression of hand motor skills after right-hemispheric application [15]. Several correlations between the intervention-induced change of motor performance of the left hand and neural processes within either hemisphere were detected in our study. In contrast, only one relevant relationship was found between changes of motor performance of the right hand and neural processes within ipsilateral M1. Neural processing underlying motor function of the right and left hand differs between hands. fMRI and PET studies show an activation within contralateral brain motor regions during a voluntary action of the right hand, whereas an activation of both hemispheres is caused by the left-hand activity [54,55]. TMS data demonstrates that an active movement of the right hand evokes greater inhibitory influence from the contralateral M1 towards the homologue area than active movement of the left hand [18,56]. It appears that right hand motor performance correlates with neural processing within the contralateral hemisphere, while left-hand motor control is closely coupled to neural processes within both the right and the left hemispheric network.

Our study did not detect differential effects between M1 and dPMC rTMS in modulation of hand motor performance. In contrast, neural processing was more frequently altered by M1 stimulation. Earlier data shows either no relevant differences between rTMS to M1 or PMC on a change of neural processing within the motor network [43], or a more powerful impact of PMC stimulation [40]. Future research should devote more attention to this relevant topic and compare effects of diverse protocols over several brain areas.

## 5. Strengths and Limitations

This is the first placebo-controlled study that directly compared the effects of rTMS over the different cortical areas of both hemispheres on motor function of either hand and the neural processing within the motor areas of either hemisphere. Our data provides additional insights into the complexity of motor processing across both hemispheres. An evident weakness of our approach is the limited number of participants, and the absence of correction for multiple testing. Another shortcoming is the missing state of art techniques of neuro-navigation to determinate the exact location of (r)TMS application.

## 6. Conclusions

The study under discussion shows that 1 Hz rTMS applied over both M1s and dPMC significantly increased CSP and ISP duration within the right hemisphere. In contrast, neural processing within the left hemisphere was not influenced. Regarding hand dexterity, both stimulation-induced improvement as well as deterioration were observed. Several correlations were found between intervention-induced changes of neural processing and hand motor performance. Changes of neural processing within either hemisphere were more frequently coupled to changes of left-hand dexterity. The relevance of these findings sheds more light upon the complex interplay within motor areas of the bi-hemispheric motor network but much remains to be further explored.

## Figures and Tables

**Figure 1 brainsci-13-00752-f001:**
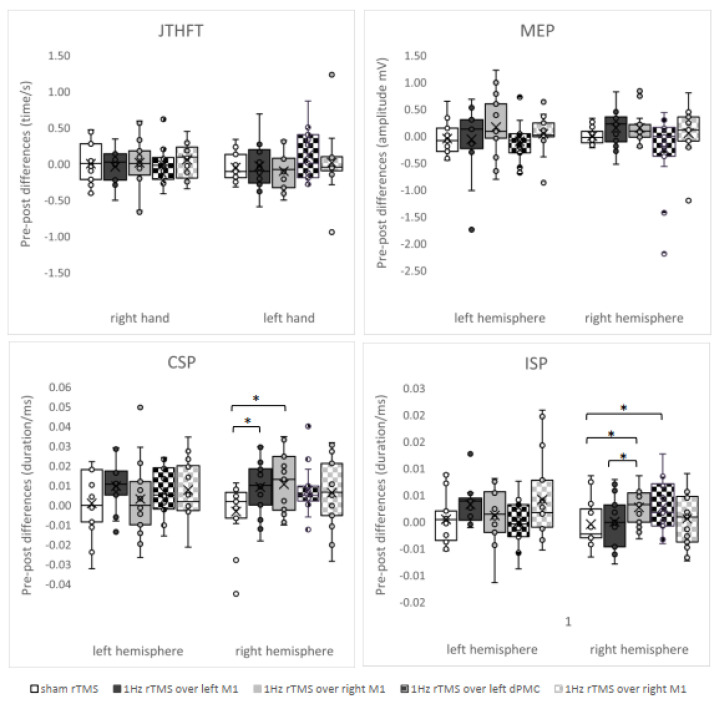
Intervention-induced changes of parameters assessed (box plot and whiskers) expressed as differences to baseline. Notes: CSP = cortical silent period; dPMC = dorsal premotor cortex; Hz = hertz; ISP = ipsilateral silent period; JTHFT = Jebsen–Taylor Hand Function Test; M1 = primary motor cortex; MEP = motor evoked potential; ms = millisecond; mV = millivolt; rTMS = repetitive transcranial magnetic stimulation; s = second. * means *p* ≤ 0.05.

**Figure 2 brainsci-13-00752-f002:**
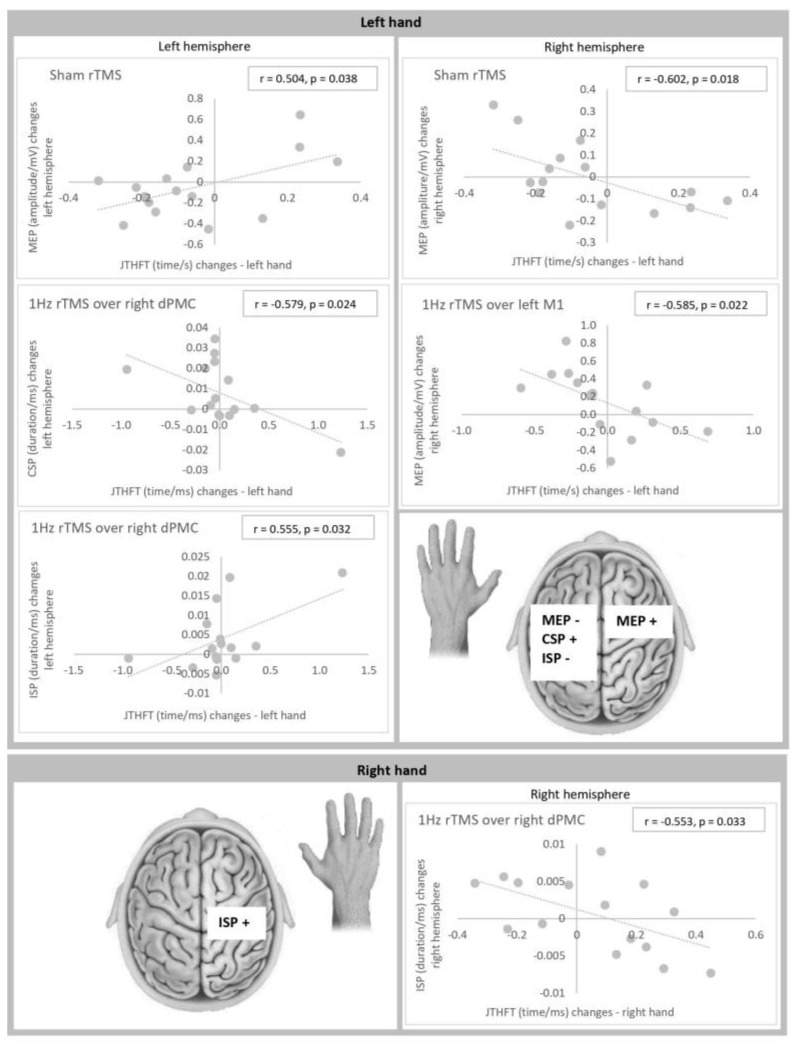
Correlations between intervention-induced changes of hand function and intervention-induced changes of electrophysiological parameters. Notes: CSP = cortical silent period; dPMC = dorsal premotor cortex; Hz = hertz; ISP = ipsilateral silent period; JTHFT = Jebsen–Taylor Hand Function Test; M1 = primary motor cortex; MEP = motor evoked potential; ms = millisecond; mV = millivolt; r = Pearson correlation; rTMS = repetitive transcranial magnetic stimulation; s = second.

**Table 1 brainsci-13-00752-t001:** Mean values and standard deviations of hand motor function test (JTHFT) and electrophysiological measures (MEP, CSP, ISP) on both time-points (pre, post).

			Placebo rTMS	1 Hz rTMS over Left M1	1 Hz rTMS over Right M1	1 Hz rTMS over Left dPMC	1 Hz rTMS over Right dPMC
JTHFT (time, s)	right hand	pre	4.31 ± 0.46	4.38 ± 0.49	4.33 ± 0.50	4.40 ± 0.54	4.38 ± 0.38
		post	4.31 ± 0.57	4.34 ± 0.48	4.34 ± 0.45	4.39 ± 0.63	4.43 ± 0.05
	left hand	pre	4.42 ± 0.53	4.54 ± 0.64	4.57 ± 0.48	4.53 ± 0.65	4.50 ± 0.39
		post	4.37 ± 0.51	4.51 ± 0.62	4.47 ± 0.43	4.65 ± 0.70	4.52 ± 0.49
MEP (size, mV)	left hemisphere	pre	0.62 ± 0.66	0.69 ± 0.85	0.69 ± 0.62	0.85 ± 0.75	0.56 ± 0.55
		post	0.57 ± 0.62	0.63 ± 0.52	0.85 ± 0.64	0.75 ± 0.60	0.60 ± 0.39
	right hemisphere	pre	0.42 ± 0.34	0.48 ± 0.34	0.29 ± 0.22 ^b^	0.64 ± 0.65	0.46 ± 0.37 ^b^
		post	0.42 ± 0.40	0.62 ± 0.37	0.43 ± 0.27	0.40 ± 0.29	0.51 ± 0.43
CSP (duration, ms)	left hemisphere	pre	0.147 ± 0.036 ^b^	0.150 ± 0.033	0.156 ± 0.038	0.156 ± 0.040	0.156 ± 0.035 ^b^
		post	0.148 ± 0.039	0.169 ± 0.039	0.159 ± 0.033	0.163 ± 0.038	0.164 ± 0.029
	right hemisphere	pre	0.160 ± 0.038	0.162 ± 0.033	0.158 ± 0.036	0.157 ± 0.043	0.162 ± 0.031
		post	0.157 ± 0.035	0.171 ± 0.030 *	0.169 ± 0.030 *	0.164 ± 0.043	0.169 ± 0.036
ISP (duration, ms)	left hemisphere	pre	0.028 ± 0.004	0.030 ± 0.003	0.030 ± 0.004	0.030 ± 0.006	0.029 ± 0.006
		post	0.028 ± 0.004	0.033 ± 0.004	0.032 ± 0.004	0.030 ± 0.005	0.033 ± 0.007
	right hemisphere	pre	0.031 ± 0.003	0.032 ± 0.004	0.030 ± 0.004	0.030 ± 0.005	0.032 ± 0.005
		post	0.030 ± 0.004	0.032 ± 0.005 ^a^	0.033 ± 0.005 *^a^	0.032 ± 0.005 *	0.032 ± 0.005

Notes: * = significant intervention-induced changes in comparison to placebo rTMS; ^a^ = significant intervention-induced differences between two real rTMS groups; ^b^ = significant differences between two groups at the baseline; CSP = cortical silent period; ISP = ipsilateral silent period; JTHFT = Jebsen–Taylor Hand Function Test; MEP = motor evoked potential; ms = millisecond; mV = millivolt; s = second.

## Data Availability

The datasets generated during and/or analyzed during the current study are available from the corresponding author on reasonable request.

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
