# Peer review of "Hemispheric Differences of 1 Hz rTMS over Motor and Premotor Cortex in Modulation of Neural Processing and Hand Function"

_brainsci, 2023, doi:10.3390/brainsci13050752_

Round 1
Reviewer 1 Report
The paper by Veldema and colleagues explores the modulation of 1Hz rTMS on motor cortex functions, demonstrating changes of CSP/ISP within the right hemisphere. The results are original and the work well presented.
I have some minor concerns. Introduction is too long and dispersive and does not focus on the main topic of the draft. I suggest to shorten this section. The same as regarding the discussion, that is abundantly descriptive.
Premotor areas include also dPMC but are not strictly related to it. Why authors did not choose to stimulate other areas such as SMA?
Authors should also discuss the role of TMS in facilitating motor function, as recently reported (10.3389/fnhum.2021.684013)
Author Response
Dear reviewer,
thank you for the opportunity to revise our manuscript “Hemispheric differences of 1Hz rTMS over motor and premotor cortex in modulation of neural processing and hand function.” (ID: brainsci-2247095).
We appreciate the interest that you have taken in our manuscript and the constructive criticism you have given. We have revised our manuscript in accordance with them and have included a point-by-point response to your comments. Changes in the manuscript are made in tracking mode.
Comment 1:
I have some minor concerns. Introduction is too long and dispersive and does not focus on the main topic of the draft. I suggest to shorten this section. The same as regarding the discussion, that is abundantly descriptive.
Answer: Thank you for this hint. We revised the introduction and the discussion in accordance with this comment.
Comment 2:
Premotor areas include also dPMC but are not strictly related to it. Why authors did not choose to stimulate other areas such as SMA?
Answer: We could demonstrate, in a previous rTMS study, that dPMC is a promising area to support hand motor performance in stroke cohorts. Thus, a more comprehensive investigation of this region as well as comparison with M1 area makes sense. We included this information in the introduction.
Comment 3:
Authors should also discuss the role of TMS in facilitating motor function, as recently reported (10.3389/fnhum.2021.684013)
Answer: The literature search using this doi reveals an article investigating the influence of intended hand movement on MEP size, in dependence on time interval before movement onset (Cirillo G, Di Vico IA, Emadi Andani M. Changes in Corticospinal Circuits During Premovement Facilitation in Physiological Conditions). We are not sure if this article was really meant?
Reviewer 2 Report
The authors aim to determine the effect of rTMS over M1 and dPMC bilaterally and the relationship with neurophysiology and hand motor function in healthy individuals. The topic is timely and of interest to researchers particularly in the fields of neurorehabilitation. This reviewer has several questions about the methods and results which will improve the readability and reproducibility of the experiment.
Introduction
· There are many words and phrases in quotation marks in section 1.1 which make it difficult to interpret the authors meaning. These phrases should be re-worded to improve clarity
· There is reference to a subchapter at the end of section 1.2 and it is unclear what is meant by subchapter
Methods
· 2.1 title – it is better to describe the people in experiments as participants and not subjects
· 2.2 – please remove the quotation marks to describe the washout period, or use a phrase that does not require quotation marks. There is a typo in the last sentence: quit is likely meant to be quiet.
· 2.5 – was a correction for multiple comparisons considered. if not, please provide a rationale
· what is the rationale for the sample size?
· how were the MEPs measured prior to statistics? how was CSP and ISP measured?
Results
· Data are plural so the first sentence should say “their data were not included.”
· how did you determine handedness of the study participants?
· Figure 1: What do the error bars indicate? standard deviation or standard error? The error bars are quite large in comparison to the means. Can the authors please comment.
· Figure 2: x- and y-axis have different scales which make it difficult to appreciate the differences and similarities between the outcomes
Discussion
· It does not appear as though the authors corrected for multiple comparisons (or provide a rationale for why they did not), so the assertion that numerous correlational relationships were found is overstated.
· The discussion is unfocused and would benefit from editing for clarity and brevity
· The discussion reads more like a review than a discussion of the authors findings in the context of the relevant literature. Description of published literature should be more succinct.
Author Response
Dear reviewer,
thank you for the opportunity to revise our manuscript “Hemispheric differences of 1Hz rTMS over motor and premotor cortex in modulation of neural processing and hand function.” (ID: brainsci-2247095).
We appreciate the interest that you have taken in our manuscript and the constructive criticism you have given. We have revised our manuscript in accordance with them and have included a point-by-point response to your comments. Changes in the manuscript are made in tracking mode.
Introduction
Comment 1:
There are many words and phrases in quotation marks in section 1.1 which make it difficult to interpret the authors meaning. These phrases should be re-worded to improve clarity.
Answer: Thank you for this comment. We removed the quotation marks and revised the section.
Comment 2:
There is reference to a subchapter at the end of section 1.2 and it is unclear what is meant by subchapter.
Answer: The Introduction chapter was substantially revised and the sentence mentioned was removed.
Methods
Comment 3:
2.1 title – it is better to describe the people in experiments as participants and not subjects
Answer: The title was revised
Comment 4:
2.2 – please remove the quotation marks to describe the washout period, or use a phrase that does not require quotation marks. There is a typo in the last sentence: quit is likely meant to be quiet.
Answer: Thank you for this hint. We removed the quotation marks and revised the typo.
Comment 5:
2.5 – was a correction for multiple comparisons considered. if not, please provide a rationale
Other statisticians recommend not doing any formal corrections for multiple comparisons when the study focuses on only a few scientifically sensible comparisons, rather than every possible comparison (the so-called planned comparison applied in our study), or when the comparisons are complementary. Some statisticians recommend never correcting for multiple comparisons while analysing data.
- No adjustments are needed for multiple comparisons. Epidemiology. 1990.
- Multiple Comparison Procedures: The Practical Solution. The American Statistician. 1990.
- Rosuvastatin to Prevent Vascular Events in Men and Women with Elevated C-Reactive Protein. N Engl J Med. 2008.
On this background, we waive on corrections for multiple comparisons.
Comment 6:
what is the rationale for the sample size?
Answer: The rationale for sample size are previous studies investigating 1Hz rTMS in modulation of neural networks (Hoogendam JM, Ramakers GM, Di Lazzaro V. Physiology of repetitive transcranial magnetic stimulation of the human brain.). An absolute majority of them included less than 15 patients and could detect significant effects on neural networks.
Comment 7:
how were the MEPs measured prior to statistics? how was CSP and ISP measured?
Answer: Section 2.3.1. Neurophysiological evaluations include the description of MEP, CSP and ISP measurements.
Results
Comment 8:
Data are plural so the first sentence should say “their data were not included.”
Answer: we revised the sentence in accordance with the comment.
Comment 9:
how did you determine handedness of the study participants?
Answer: Hand dominance was tested by a self-report questionnaire. We included this information in the main text and the reference list.
Comment 10:
Figure 1: What do the error bars indicate? standard deviation or standard error? The error bars are quite large in comparison to the means. Can the authors please comment.
Answer: There are standard deviations. This information was supplied in the figure description. The discussion of SD can be found on the line 268-273.
Comment 11:
Figure 2: x- and y-axis have different scales which make it difficult to appreciate the differences and similarities between the outcomes
Answer: The figures illustrate the correlation between the changes of hand motor function and the neurophysiological changes measured by different tests (MEP, CSP, ISP). This leads necessary to differential scales.
Discussion
Comment 12:
It does not appear as though the authors corrected for multiple comparisons (or provide a rationale for why they did not), so the assertion that numerous correlational relationships were found is overstated.
Answer: Other statisticians recommend not doing any formal corrections for multiple comparisons when the study focuses on only a few scientifically sensible comparisons, rather than every possible comparison (the so-called planned comparison applied in our study), or when the comparisons are complementary. Some statisticians recommend never correcting for multiple comparisons while analysing data.
- No adjustments are needed for multiple comparisons. Epidemiology. 1990.
- Multiple Comparison Procedures: The Practical Solution. The American Statistician. 1990.
- Rosuvastatin to Prevent Vascular Events in Men and Women with Elevated C-Reactive Protein. N Engl J Med. 2008.
On this background, we waive on corrections for multiple comparisons.
Comment 13:
The discussion is unfocused and would benefit from editing for clarity and brevity
The discussion reads more like a review than a discussion of the authors findings in the context of the relevant literature. Description of published literature should be more succinct.
Answer: Thank you for this hint. We revised the discussion substantially and reduced its length.
Reviewer 3 Report
I think is a good job. Interesting and interesting for research.
Author Response
Dear reviewer,
thank you very much for your positive evaluation of our manuscript.
Reviewer 4 Report
It was a pleasure to evaluate the article entitled “Hemispheric Differences of 1Hz rTMS over Motor and Premotor Cortex in Modulation of Neural Processing and Hand Function”. The authors' goal was to answer the important question of the variability between the left and right hemispheres after 1 Hz rTMS intervention. The authors tested four different targets and added additional control. In general, the experiments were performed with a high methodological standard in terms of their approach, but the article’s text requires some improvements.
While the use of the English language is generally on par with that of a native speaker, there are some minor mistakes in grammar, such as improper preposition use (“prior” vs “prior to”), verb plurality mistakes (“lengthening were”), and syntax (“improved also”). However, these mistakes generally do not disqualify the article from being readable, aside from the lengthy nature of the text itself. Carefully revising the entire paper should yield the necessary corrections.
Language aside, I have some major recommendations for the following parts of the manuscript:
- The authors described the results from many studies in detail, but without a clear conclusion or logic behind why these studies are relevant to the current work. Moreover, many of these studies are not directly connected to the methods or results presented by the authors at all. I understand that some of the third-party funding may bolster and explain the results of the current research; however, I found several parts of the introduction to be overly bulky and unnecessary. These parts should be shortened and polished:
- Lines 42-55: The authors failed to adjust both the number of pulses per rTMS intervention and the estimation of the post-rTMS effect (testing the effect at different time intervals after rTMS) in the current study. Therefore, I see no reason why this information should be included in the introduction and there’s certainly no reason to open the article with it.
- Lines 61—64: The authors described the effect of different rTMS interventions in detail (1Hz, 10Hz, 15Hz, 20Hz, iTBS, cTBS), but only the 1 Hz protocol was applied in the current manuscript.
- Lines 105-109: Presenting the number of articles that indicate the application of rTMS over different brain areas is absolutely noninformative because it does not give the reader any information about the results of these experiments. Additionally, the authors cannot guarantee that they have collected every piece of research that performed stimulation over these areas.
In conclusion, I recommend decreasing the size of the introduction by at least a factor of two. The goal should be to focus on the data that allows the reader to understand the reason for conducting the current research.
- Controlling the location of the coil during the experiment and switching between targets. I could not find information that suggests the authors were using navigation TMS in their experiments. If navigation TMS was not used, then I am concerned about how the TMS coil was controlled over the M1 and dPMC areas, and how the switching was performed between stimulation areas. The corresponding section should be added to the “Limitations” section.
- In the Methods section, I could not find any description of how the authors measured maximal contractions, and correspondingly, how they evaluated the different percentages for the tasks. This should be included in the “Methods” section.
- The “sham” control was applied incorrectly. In my humble opinion, the application of 0% intensity cannot be classified as a sham control. Specifically, TMS induces auditory (clicking), vibrational, and electrical (activation of the cranial muscles) artifacts. However, applying 0% intensity induces none of the above-mentioned effects. This kind of control can be classified as a “placebo”, but not as a “sham”.
- The data representation on Figure 1 should be improved. Firstly, the author did not describe the normalization procedure for the presented data. Secondly, I recommend using a box plot instead of a bar chart. A box plot will enable the easy recognition of the data distribution and statistical significance. This change will be especially valuable for the CSP and ISP graphs. Additionally, I am wondering if the authors presented all the statistically significant values in the plot.
- Figure 1. MEP. I would prefer to see the responses for pre- and post-stimulation in the same graph. Specifically, I suggest not subtracting the pre-values from the post-values, but rather plotting them all together as a supplementary figure. The deviation of the values before intervention will help estimate the actual changes in the parameters after the intervention. Please use the box plot function (my comment 5).
- The Discussion section requires similar improvements as mentioned above concerning the Introduction. I found a lot of examples and discussions that are not related to the current study. Meanwhile, the discussion of the primary results, specifically the effect of the 1 Hz intervention at different locations, is hidden in the lengthy and noninformative text.
I found these 7 points to be the most critical. I would be happy to evaluate the article again after the authors improve the text according to my recommendations.
In addition to the major issues mentioned above, I have some minor points that I would like to address:
Line 296: “in this young and middle-aged cohort”: I do not understand how the authors divided or estimated their population of 15 subjects as a “young and middle-aged” group. I suggest either clearly comparing the results with articles that use groups of the same age or avoiding the classification of the subject population’s age from the current article.
Line 327: “The effects of rTMS on corticospinal excitability have been widely investigated and were summarized in systematic reviews [1,10,11].” This sentence is vague and does not include valuable information. I suggest rewriting it.
Line 341 “Regarding 1Hz rTMS over dPMC, left hemispheric subthreshold intensities and their 341 influence on ipsilateral M1 were tested [1,39-43].” This is one more example of a noninformative sentence, which sheds light on the reason why I’ve stated that the article should be rewritten to increase readability.
Line 259 “Correlations”: Why was JTFHT chosen as a primary parameter to correlate with all other results instead of any other? Can the authors present reasons as to why there are physiological advantages to this kind of comparison?
Line 479 “effects of rTMS”: Since only 1 Hz intervention was applied, I recommend being more specific and clearly mentioning it in the conclusion.
Table 1: I would like to see the entire table on one page, and also suggest adding spaces or horizontal lines between the different parameters.
Figure 1: The legend of the “MEP” subplot does not really indicate the difference between sham and 1 Hz rTMS over left dPMC.

Author Response
Dear reviewer,
thank you for the opportunity to revise our manuscript “Hemispheric differences of 1Hz rTMS over motor and premotor cortex in modulation of neural processing and hand function.” (ID: brainsci-2247095).
We appreciate the interest that you have taken in our manuscript and the constructive criticism you have given. We have revised our manuscript in accordance with them and have included a point-by-point response to your comments. Changes in the manuscript are made in tracking mode.
Comment 1:
While the use of the English language is generally on par with that of a native speaker, there are some minor mistakes in grammar, such as improper preposition use (“prior” vs “prior to”), verb plurality mistakes (“lengthening were”), and syntax (“improved also”). However, these mistakes generally do not disqualify the article from being readable, aside from the lengthy nature of the text itself. Carefully revising the entire paper should yield the necessary corrections.
Answer: Thank you for this hint. We checked the manuscript regarding grammar errors.
Language aside, I have some major recommendations for the following parts of the manuscript:
Comment 2:
The authors described the results from many studies in detail, but without a clear conclusion or logic behind why these studies are relevant to the current work. Moreover, many of these studies are not directly connected to the methods or results presented by the authors at all. I understand that some of the third-party funding may bolster and explain the results of the current research; however, I found several parts of the introduction to be overly bulky and unnecessary. These parts should be shortened and polished:
Answer: Thank you for this hint. A comprehensive revision of both, Introduction and Discussion was performed.
Comment 3:
Lines 42-55: The authors failed to adjust both the number of pulses per rTMS intervention and the estimation of the post-rTMS effect (testing the effect at different time intervals after rTMS) in the current study. Therefore, I see no reason why this information should be included in the introduction and there’s certainly no reason to open the article with it.
Answer: We revised the chapter in accordance with this comment.
Lines 61—64: The authors described the effect of different rTMS interventions in detail (1Hz, 10Hz, 15Hz, 20Hz, iTBS, cTBS), but only the 1 Hz protocol was applied in the current manuscript.
Answer: The text was revised in accordance
Comment 4:
Lines 105-109: Presenting the number of articles that indicate the application of rTMS over different brain areas is absolutely noninformative because it does not give the reader any information about the results of these experiments. Additionally, the authors cannot guarantee that they have collected every piece of research that performed stimulation over these areas.
Answer: We revised the chapter.
Comment 5:
In conclusion, I recommend decreasing the size of the introduction by at least a factor of two. The goal should be to focus on the data that allows the reader to understand the reason for conducting the current research.
Answer: We revised the manuscript in accordance with the comment.
Comment 6:
Controlling the location of the coil during the experiment and switching between targets. I could not find information that suggests the authors were using navigation TMS in their experiments. If navigation TMS was not used, then I am concerned about how the TMS coil was controlled over the M1 and dPMC areas, and how the switching was performed between stimulation areas. The corresponding section should be added to the “Limitations” section.
Answer: We supplemented the description of coil placement control in the subchapter intervention. The absence of TMS-navigation has already been mentioned in the chapter Limitations.
Comment 7:
In the Methods section, I could not find any description of how the authors measured maximal contractions, and correspondingly, how they evaluated the different percentages for the tasks. This should be included in the “Methods” section.
Answer: The force-related measurements were controlled using force transducer. This has already been described within the main text for both, CSP (line 189-199) and ISP (line 196-197).
Comment 8:
The “sham” control was applied incorrectly. In my humble opinion, the application of 0% intensity cannot be classified as a sham control. Specifically, TMS induces auditory (clicking), vibrational, and electrical (activation of the cranial muscles) artifacts. However, applying 0% intensity induces none of the above-mentioned effects. This kind of control can be classified as a “placebo”, but not as a “sham”.
Answer: To our knowledge “sham” and “placebo” are used as a synonym. We also could not detect a clear distinction between both terms in the available literature. The application of 0% intensity is a common form of placebo/sham in rTMS studies.
Comment 9:
The data representation on Figure 1 should be improved. Firstly, the author did not describe the normalization procedure for the presented data. Secondly, I recommend using a box plot instead of a bar chart. A box plot will enable the easy recognition of the data distribution and statistical significance. This change will be especially valuable for the CSP and ISP graphs. Additionally, I am wondering if the authors presented all the statistically significant values in the plot.
Answer: Thank you for the suggestion. However, we found that a bar chart is an appropriate method to present results of this study. We do not understand what exactly is meant by “describe the normalization procedure for the presented data”.
Comment 10:
Figure 1. MEP. I would prefer to see the responses for pre- and post-stimulation in the same graph. Specifically, I suggest not subtracting the pre-values from the post-values, but rather plotting them all together as a supplementary figure. The deviation of the values before intervention will help estimate the actual changes in the parameters after the intervention. Please use the box plot function (my comment 5).
Answer: The pre and post values (Mean and SD) for all assessments are presented in Table 1. Figure 1 presents the intervention-induced changes (Mean and SD). In our opinion, this is an appropriate method to present our findings.
Comment 11:
The Discussion section requires similar improvements as mentioned above concerning the Introduction. I found a lot of examples and discussions that are not related to the current study. Meanwhile, the discussion of the primary results, specifically the effect of the 1 Hz intervention at different locations, is hidden in the lengthy and noninformative text.
Answer: Thank you for this comment. We revised Discussion in accordance with them.
In addition to the major issues mentioned above, I have some minor points that I would like to address:
Comment 12:
Line 296: “in this young and middle-aged cohort”: I do not understand how the authors divided or estimated their population of 15 subjects as a “young and middle-aged” group. I suggest either clearly comparing the results with articles that use groups of the same age or avoiding the classification of the subject population’s age from the current article.
Answer: Thank you for this hint. We revised the discussion in accordance with them.
Comment 13:
Line 327: “The effects of rTMS on corticospinal excitability have been widely investigated and were summarized in systematic reviews [1,10,11].” This sentence is vague and does not include valuable information. I suggest rewriting it.
Answer: The sentence was removed.
Comment 14:
Line 341 “Regarding 1Hz rTMS over dPMC, left hemispheric subthreshold intensities and their 341 influence on ipsilateral M1 were tested [1,39-43].” This is one more example of a noninformative sentence, which sheds light on the reason why I’ve stated that the article should be rewritten to increase readability.
Answer: We revised the text.
Comment 15:
Line 259 “Correlations”: Why was JTFHT chosen as a primary parameter to correlate with all other results instead of any other? Can the authors present reasons as to why there are physiological advantages to this kind of comparison?
Answer: This study evaluates the impact of rTMS on motor function of both hands (evaluated by JTHFT) and the neural processing within both hemispheres (evaluated by MEP, CSP and ISP). The correlations between motor (JTHFT) and electrophysiological (MEP, CSP and ISP) changes should improve the understanding of the neural background of TMS-induced behavioural effects.
Comment 16:
Line 479 “effects of rTMS”: Since only 1 Hz intervention was applied, I recommend being more specific and clearly mentioning it in the conclusion.
Answer: Unfortunately, we cannot find the information mentioned on line 479. However, the discussion chapter was extensively revised. We hope that your comment is thereby considered.
Comment 17:
Table 1: I would like to see the entire table on one page, and also suggest adding spaces or horizontal lines between the different parameters.
Answer: The final positioning and outline of the Tables lies in the hands of editorial team.
Comment 18:
Figure 1: The legend of the “MEP” subplot does not really indicate the difference between sham and 1 Hz rTMS over left dPMC.
Answer: Thank you for this hint. The submitted data are correct. A slightly larger Figure should enhance the readability. This lies in the hands of the editorial team.
Round 2
Reviewer 4 Report
I would like to thank the authors for doing great job on the improvement of the manuscript. I am glad that my comments proved useful and could aid in increasing the quality of the work. However, I found several critical points that were not appropriately addressed, which consequently led to my recommendation to reject the article.
Namely, my main concerns relate to the use of 0% intensity as a sham control (Comment 8). The authors insisted that this is a commonly used practice and that 0% intensity could be considered a sham control. I could not find strong support for this statement. Additionally, my experience and knowledge in the subject matter prevent me from agreeing with authors’ conclusion. As I mentioned in the original review, “TMS induces auditory (clicking), vibrational, and electrical (activation of the cranial muscles) artifacts. However, applying 0% intensity induces none of the above-mentioned effects.” The authors applied stimulation outside of the motor cortex (premotor cortex), and therefore, a realistic sham is required to control the observed effect. I recommend that the authors become acquainted with the following fundamental article in this field (doi: 10.1016/j.neuroimage.2018.10.052). Even in this case, the primary measurements were performed through the application of EMG, but not EEG methods.
All rTMS protocols (including 1 Hz) induce a weaker effect. The MEP amplitude variability is higher compared to the inhibition/facilitation effect induced by rTMS (10.1016/j.brs.2021.05.013). The group population studies (doi.org/10.1016/j.cortex.2021.02.024; doi.org/10.1016/j.clinph.2019.11.002) report high inter- and intra-subject variability, and a near absence of the effect in the case of a large population. Therefore, the effect of rTMS on the MEP should be considered carefully and with maximum controls. Such controls are absent in the presented article, which is why I recommend carefully scrutinizing the presented results.
My comment 6: I could not find the subchapter “Intervention”, and the authors did not clearly explain where I could find the answer to my comment. Therefore, I must interpret this as a failure to answer my question.
My comment 7: I read the information presented on lines 186-199 of the original version of the manuscript, as well as the revised version after a reply to my comment was proved. Unfortunately, I did not find the answer on my original question.
My final and no less important concern is related to my comment 9, where I stated that I would like to see Figure 1 visualized as a box plot. I consider the response “We found that a bar chart is an appropriate method to present results of this study” completely unsatisfactory due to the fact that a box plot is a common and most reliable method for the visual representation of MEP amplitude (original, and normalized). The refusal to present this form of data visualization, at the very least as supplementary material, as well as the absence of a strong argument against the use of this form of data visualization leads to my distrust in the presented data overall.
